# The Monocyte, a Maestro in the Tumor Microenvironment (TME) of Breast Cancer

**DOI:** 10.3390/cancers14215460

**Published:** 2022-11-07

**Authors:** Hoda T. Amer, Ulrike Stein, Hend M. El Tayebi

**Affiliations:** 1Molecular Pharmacology Research Group, Department of Pharmacology and Toxicology, Faculty of Pharmacy and Biotechnology, German University in Cairo, Cairo 11865, Egypt; 2Translational Oncology of Solid Tumors, Experimental and Clinical Research Center, Charité—Universitäsmedizin Berlin and Max-Delbrük-Center for Molecular Medicine in the Helmholtz Association, 10117 Berlin, Germany; 3German Cancer Consortium (DKTK), 69120 Heidelberg, Germany

**Keywords:** monocytes, tumor microenvironment, tumor-associated macrophages, breast cancer, TNBC, monocyte-derived populations, IL-10

## Abstract

**Simple Summary:**

Breast cancer is one of the most prevalent cancers worldwide, surpassing lung cancer as the leading cause of overall cancer incidence. Available possible treatments nowadays include chemotherapy, hormonal therapy, and HER2-targeted therapy. Chemotherapy is notorious for its severe adverse effects. On the other hand, hormonal and HER2-targeted therapies only cover a narrow range of breast cancer subtypes. Accordingly, it is important to shed light on other therapy options. For this reason, immunotherapy nowadays is one of the most important research topics. It can be accomplished either by enhancing the pro-inflammatory immunity or suppressing the anti-inflammatory immunity. This review article aims to shed light on the importance of monocytes in the TME of breast cancer. The review also aims to highlight the behavior of the monocyte-derived populations, especially the anti-inflammatory populations. Thus, suppressing this anti-inflammatory activity might have a remarkable impact on future immunotherapy research.

**Abstract:**

Breast cancer (BC) is well-known for being a leading cause of death worldwide. It is classified molecularly into luminal A, luminal B HER2−, luminal B HER2+, HER2+, and triple-negative breast cancer (TNBC). These subtypes differ in their prognosis; thus, understanding the tumor microenvironment (TME) makes new treatment strategies possible. The TME contains populations that exhibit anti-tumorigenic actions such as tumor-associated eosinophils. Moreover, it contains pro-tumorigenic populations such as tumor-associated neutrophils (TANs), or monocyte-derived populations. The monocyte-derived populations are tumor-associated macrophages (TAMs) and MDSCs. Thus, a monocyte can be considered a maestro within the TME. Moreover, the expansion of monocytes in the TME depends on many factors such as the BC stage, the presence of macrophage colony-stimulating factor (M-CSF), and the presence of some chemoattractants. After expansion, monocytes can differentiate into pro-inflammatory populations such as M1 macrophages or anti-inflammatory populations such as M2 macrophages according to the nature of cytokines present in the TME. Differentiation to TAMs depends on various factors such as the BC subtype, the presence of anti-inflammatory cytokines, and epigenetic factors. Furthermore, TAMs and MDSCs not only have a role in tumor progression but also are key players in metastasis. Thus, understanding the monocytes further can introduce new target therapies.

## 1. Introduction

### 1.1. Introduction to BC

Breast cancer (BC) is one of the most commonly diagnosed cancers in women. It is not only a leading cause of cancer-related death, but also a leading cause of disease-associated death among women worldwide [1]. The number of death cases due to BC was 685,000 in 2020, making it the fifth leading cause of cancer mortality worldwide [2]. Although surgeries or chemotherapies are chances of recovery, the recovery chance decreases as cancer progresses [3]. The prognosis of BC depends on many factors, including traditional clinicopathological variables such as tumor grade, tumor size, and nodal involvement, and it has been shown to possess distinct behavior according to each molecular subtype. It was noted that each molecular subtype has different histopathological and biological features which lead to different and unique treatment responses and strategies [4]. 

The subtypes of BC are divided molecularly according to the expression of estrogen receptor (ER), progesterone receptor (PR), and human epidermal growth factor receptor 2 (HER2) with the percentage of the proliferating index (Ki67). With this in mind, BC is classified accordingly into five different subtypes: luminal A, luminal B HER2-negative, luminal B HER2-positive, HER2-positive, and triple-negative BC (TNBC) (basal-like) [5]. Luminal A subtype is characterized by expression of ER and/or PR with no expression of HER2 and with Ki67 < 14%. Moreover, the luminal A subtype expresses BCL-2 cytokeratin CK 8/18 and GATA 3 marker, which is expressed with its highest levels in this subtype [6]. Luminal A is considered the most prevalent with a prevalence of 50–60% among BC patients and is characterized by the best prognosis with a relapse rate of 27.8% [7]. Luminal B abundance is 10–20% among BC patients; it has a more aggressive diagnostic profile than luminal A and a worse prognosis. In addition, it has about 30% bone recurrency and 13.8% liver recurrency [6]. Despite the possible treatment of luminal B with tamoxifen, it responds more to chemotherapy [6]. Luminal B is further classified into two subgroups: luminal B HER2-negative subtype and luminal B HER2-positive. Luminal B HER2-negative is characterized by the expression of ER and/or PR with no expression of HER2 and with Ki67 ≥ 14%. Luminal B HER2-positive is ER- and/or PR-positive and HER2-positive with the expression of any Ki67 percentage [8]. Furthermore, the HER2-positive (enriched) subtype is hormonal-receptor-negative, and only HER2 receptors are expressed with any Ki67 percentage. The HER2-enriched subtype suffers from a worse prognosis than luminal subtypes, although anti-HER2 treatment, e.g., trastuzumab, is possible. It has lower survival rates (10-year survival rate of 12%) in comparison to luminal subgroups (10-year survival rates of 50–55%) [9]. Finally, TNBC (basal-like) is ER-, PR-, and HER2-negative. The TNBC subtype expresses cytokeratins (e.g., CK5, CK17) and epidermal growth factor receptor (EGFR) [10]. This subtype is characterized by high p53 and BRCA1 mutations. BRCA1 is very important and critical in DNA repair [11]. The TNBC subtype is more common in early age, African origin, and increased tumor size with node involvement. It has the highest relapse rate of all subgroups, and relapse usually happens in the first 3 years [12]. Some papers classify the TNBC as not only basal-like but also normal-breast-like and claudin-low. Normal-breast-like is also TNBC but not basal-like as it does not express CK5 and EGFR [13]. The claudin-low subtype is characterized by low expression of genes involved in intercellular adhesion and tight junctions, including claudin-3, -4, and -7; cingulin; occludin; and E-cadherin, and therefore the name of this subtype is claudin-low [14]. This subtype is usually positioned in the hierarchical clustering near the basal-like tumor. Thus, both subtypes share some common gene expression characteristics such as low expression of both HER2 and hormonal receptors, as mentioned before. In contrast to the basal-like subtype, this subtype overexpresses a set group of 40 genes that are related to the immune response, resulting in a high infiltration of tumor immune system cells [14]. The claudin-low subtype has a poor prognosis with a very rare subset of tumors (12–14%) and with high-grade infiltrating ductal carcinomas [14]. Not all claudin-low tumors are negative for hormone receptors (TNBC), as 20% of claudin-low tumors were found to be positive for hormone receptors. Claudin-low tumors in general show poor response to neoadjuvant chemotherapy with relatively intermediate prognosis values between basal and luminal tumors [15].

Generally, gene expression profiling has become a useful tool for breast cancer classification treatment. Although, as previously mentioned, the treatment of HER2-enriched and TNBC is well-defined for anti-HER2 and chemotherapy treatment, respectively, the hormonal subgroup still faces a clinical challenge. Briefly, all luminal tumors are candidates for anti-hormonal therapy. However, unexpectedly, some tumors within the luminal subclass have a more proliferative profile with poorer outcomes and thus are considered for additional therapy. In this context, the common classification which is only based on the molecular intrinsic subtypes as proposed in some papers might not be enough as it only divides the luminal tumors into the luminal A subtype and luminal B. Indeed, this classification is not sufficient for clinical decisions because the luminal tumors present a prognostic range rather than an exact clinical outcome for either group. Accordingly, in a recent study, the RNA-Seq expression profiles split the luminal A samples into two subgroups, namely LumA-R1 and LumA-R2. The lobular-enriched LumA-R2 sample group is characterized by a distinct gene overexpression pattern. This category was associated with significantly reduced recurrence risk compared with the more proliferative LumA-R1 subgroup. Interestingly and most importantly, overexpressed genes were significantly enriched for functions related to the immune system, including genes of upstream T-cell-receptor signaling pathways. The study concluded that the elevated mRNA levels of immune-related genes in LumA-R2 samples indicate increased levels of infiltration of immune system cells into the tumors [16].

Another study investigated the link between neurogenesis, tertiary lymphoid structures, plasma cells, and B lymphocytes in different samples of the basal-like subtype. Accordingly, the study further categorized the basal-like tumors into two subgroups, namely C2 and C3. These two basal-like enriched clusters showed a major biological discrepancy relative to immune response. Briefly, this discrepancy was characterized by a decreasing anti-tumorigenic immune gradient from C3 to C2. Additionally, high neurogenesis activity was found for C2 tumors. Furthermore, the upregulation of immune checkpoints characterized C3 more than C2. However, VTCN1 (B7-H4) was upregulated in C2 more than in the C3 profile [17]. VTZN1 gene codes for a B7 immunoregulatory protein, which possesses an immunosuppressive activity through the inhibition of T-cell activation and clonal expansion [18]. In ovarian carcinoma and glioma, it was observed that macrophages expressing VTCN1 have been directly correlated to the inhibition of T-cell immune response [19]. Accordingly, VTNC1 might actively participate in the C2 immunosuppressive phenotype. It is worth mentioning that the study suggested that tumor-associated macrophages (discussed later on in this review) are crucial actors of tumor fate and therefore represent promising immunotherapeutic targets [20]. Consequently, numerous macrophage-directed therapeutic approaches are under investigation and should be considered in the C2 subtype [17].

### 1.2. Introduction to the Tumor Microenvironment (TME)

Tumors have the ability to recruit stromal cells (e.g., fibroblasts), immune cells, and vascular cells through the secretion of growth factors, cytokines, and chemokines. Consequently, tumors build a tumor microenvironment (TME) by releasing growth-promoting signals as well as remodeling tissue structure affecting initiation, progression, metastasis, vascularization, and therapy responses [17]. Many treatments focus only on the cancer cell itself and ignore the TME, which is actually a key player in BC progression and development [21]. Tumors not only try to escape from the host immune system but also benefit from the infiltrating cells by modifying their functions to create a microenvironment that is favorable to its progression [22]. Within the TME, different stromal, immune, and regulatory cells may stimulate or inhibit tumor growth. For example, fibroblasts that represent the majority of stromal cells present in the TME inhibit the early stages of tumor progression. This inhibition happens through the production of various fibroblast factors and IL-6. However, it was reported that cancer cells have been associated with the alteration of fibroblasts into cancer-associated fibroblasts (CAFs) [23]. CAFs secrete various growth factors and cytokines, including fibroblast growth factor, human growth factor, tenascin, thrombospondin-1, TGF-β, and stromal cell-derived factor 1 SDF-1 or CXCL12. These factors promote BC proliferation and metastasis. Moreover, TGF-β and PDGF are secreted by tumor cells, causing the migration of fibroblasts to the TME and initiating their transdifferentiation to CAFs. Basically, fibroblasts are well known to be attracted to the wound site, where they undergo fibroblast-to-myofibroblast transdifferentiation under the impact of some platelet-derived cytokines, namely TGFβ1 and PDGF. The presence of TGFβ1 and PDGF at the wound site guarantees a proper wound-healing process through the maintenance of an activated myofibroblast network. The recruitment of the myofibroblast cohort during wound healing is mimicked by most solid cancers during growth and migration. In the presence of tumors, TGF-β1 and PDGF production is accomplished by the cancer cells [24]. CAFs are also known to promote angiogenesis and remodel the extracellular matrix (ECM) [23]. In addition to CAFs, neutrophils are abundantly found not only in human blood but also within the TME and have either pro- or anti-tumorigenic properties. According to cytokines in the TME, tumor-associated neutrophils (TANs) are polarized either into pro-inflammatory/anti-tumorigenic (N1 phenotype) or anti-inflammatory/pro-tumorigenic (N2 phenotype). Moreover, their migration from the blood circulation into the TME is triggered by IL-8 (CXCL8-CXCR1/2 axis) expressed by tumor cells [25]. Normally, neutrophils do not secrete oncostatin M. However, it was found that on interaction with cancer cells, oncostatin M becomes highly expressed in TANs. Commonly, oncostatin M has an inhibitory effect on cell proliferation of BC cell lines having a pro-inflammatory response by inducing chemotaxis and adhesion of neutrophils [23]. Despite its pro-inflammatory response, oncostatin M has also been shown to promote tumor progression by enhancing angiogenesis and metastasis in BC cell lines (MDA-MB-23 and T47D) [26], suggesting that TANs predominate in TNBC in comparison to non-TNBC.

Besides CAFs and TANs, tumor-associated eosinophils play a role in TME. Eosinophils are well known for parasitic and bacterial infections and inflammatory diseases such as allergic asthma and chronic obstructive pulmonary disease, in addition to the high secretion of IL-5 and other eosinophilia granules [27]. However, recent data showed that in the human breast, eosinophils’ presence is critical for mammary gland development. Of interest, eosinophils have been observed at the edge of BC biopsy wounds; this finding suggests that BC biopsies can trigger the recruitment of eosinophils and other inflammatory cells [28]. Basically, a low eosinophil count in the peripheral blood of BC patients is considered a major risk factor for BC relapse. These data suggest a positive correlation between improved prognosis and tumor-associated eosinophils [28]. It was noted that eosinophils were not present within the TME of BC but were present in the TME of other cancers such as colon and lung. In Hodgkin’s lymphoma, their presence in the TME indicates a poor prognosis [29]. Further studies must be conducted to determine whether tumor-associated eosinophils contribute to immune suppression or immune stimulation within the TME. Additionally, a lack of tumor-associated eosinophils was observed in the invasive BC TME [30]. In addition to the previously mentioned immune cells, monocytes play a very critical role in the TME either by themselves or by polarization to different cells such as dendritic cells, myeloid-derived suppressor cells (MDSCs), and macrophages. In this review, we shed light on the importance of the monocyte in the TME, focusing on its nature and factors affecting its expansion and differentiation. In addition, our review extensively discusses the different phenotypes of monocyte-derived macrophages, especially tumor-associated macrophages (TAMs), addressing their functionality and anti-inflammatory activity in BC.

## 2. Methods

The research was performed at the States National Library of Medicine (PubMed). “Breast Cancer”, “Tumor Micro-environment”, “Monocytes”, “Tumor-associated Macrophages (TAMs)”, and “Triple Negative Breast Cancer” were the descriptors used during the search process. The identified records from the search process included original research papers, book chapters, and reviews. These records were screened according to their relevance to the aim of the review and summarized. Research studies that discussed monocyte behavior with no correlation to monocyte expansion or differentiation were excluded. Additionally, papers that focused on TAMs in any cancer type other than breast cancer were excluded, as were papers that focused on monocytes in any disease other than cancer. Inclusion criteria were complete English publications; papers discussing the nature of monocytes in the TME; papers focusing on the immune populations within the tumor of BC; and papers highlighting the behavior of monocyte-derived populations, especially TAMs. with detailed information about contributors, methods, and analyzed results. Data collection was performed during July and August 2022.

## 3. Results

### 3.1. Monocytes’ Classification in the TME of BC

Human peripheral blood monocytes are the type of immune cells arising from the myeloid lineage [31]. Monocytes are divided into three subsets based on the expression of CD16 and CD14 surface markers [32]. The CD16 (FcgRIII) molecule was only known initially to be expressed on mature macrophages; however, it has been recognized recently as a surface marker for monocytes [33]. The three subsets of monocytes are “classical” CD14+CD16− monocytes which make up around 85% of monocytes, “intermediate” CD14+CD16+ monocytes which account for 5–10% of total monocytes, and finally “non-classical” CD14-CD16+ monocytes which also account for about 5–10% of monocytes [32].

### 3.2. Monocytes and Diagnosis

A study group used receiver operating characteristic (ROC) curve analysis to determine the practicability of CD14+CD16+ monocytes as a BC diagnostic indicator. They compared the BC patients with healthy donors. The results showed that most monocytes were observed to express CD14 intensely and did not express CD16 in either BC patients or controls (CD14+CD16+ monocytes) and that the levels were not significantly different between both groups. Only a minor population of monocytes co-expressed CD16 and CD14 (CD14+CD16+ monocytes), and this population was shown to be more abundant in BC patients than in controls [33]. Thus, CD14+CD16+ monocytes can have a diagnostic role in BC progression.

### 3.3. Monocytes’ Expansion in the TME

During our research, we found various studies in the literature addressing the factors that affect the expansion of monocytes in the TME. Accordingly, this review categorizes them collectively into two major groups: cytokine-related factors and non-cytokine-related factors.

#### 3.3.1. (A) Cytokine-Related Factors

##### Macrophage Colony-Stimulating Factor (M-CSF)

M-CSF is a secreted cytokine that has many functions influencing hematopoietic stem cells, differentiating them into macrophages or other related cell types. Besides its function in differentiation, it is also involved in the survival and expansion of both monocytes and bone marrow progenitor cells [1]. Thus, monocyte expansion can be induced in vitro using M-CSF.

##### CXCL16

Fibroblasts were found to induce the secretion of CXCL16 upon cell cultivation under TNBC/monocyte conditions. CXCL16 is considered a chemoattractant for T cells and bone marrow-derived fibroblast precursors. Moreover, it also acts as a key player in fibrosis and myofibroblast activation in renal fibrosis. Interestingly, it is also recognized as a myeloid cell chemoattractant [31]. Human cancer-associated fibroblasts (CAFs) were isolated from both ER- breast tumors and ER+ breast tumors. They were then cultured in vitro, and their supernatants were collected. Then, angiogenesis protein assay, ELISA, and protein microarray assays were performed. On analyzing the results, three proteins were found elevated: CXCL16, amphiregulin (a tumor progression protein), and tissue inhibitor metalloproteinases (TIMP1) [34]. These three proteins were found to be overexpressed in ER- but not in ER+ breast cancer CAFs. Thus, CAFs isolated from TNBC tumors express CXCL16, which plays an important role in the recruitment of myeloid cells, including monocytes and fibroblasts. It is important to highlight that the CXCL16 expression by fibroblasts was found to vary among TN breast tumors. This finding may be due to the heterogeneity between the different subgroups of TNBC, as discussed previously in this review. These data shed light on the importance of studying the tumor stroma of TNBC since it can contain some attractive cytokines which recruit immune cells more in comparison to the tumor stroma of luminal subtypes [31]. It is of great interest to know that also GM-CSF and CXCL4 are important for fibroblast recruitment or activation [35], making GM-CSF not only directly but also indirectly important for the expansion of monocytes. The indirect impact happens through the recruitment of CAFs that in turn produce CXCL16, recruiting more and more monocytes in hormone-independent BC.

##### YKL-39 as a Monocyte Attracting Factor Produced by TAMs 

Macrophages produce chitinase-like proteins (CLPs), including chitinase 3-like 1, CHI3L1 (YKL-40), chitinase 3-like 2 (YKL-39), CHI3L2, stabilin-1-interacting chitinase-like protein (SI-CLP), and chitinase-like 3/4, CHI3L3/4 (YM1/2). YKL-40 is the best-investigated CLP that combines both pro-inflammatory and pro-angiogenic properties. It was found that elevated levels of YKL-40 correlate with metastasis and poor overall survival in many types of human cancers, including BC. YKL-39 is a very similar homolog to YKL-40 and was usually observed as a secreted protein in the primary culture of human articular chondrocytes [36].

However, it was discovered that YKL-39 is expressed in TAMs in human BC. TGF-β was found to be the major stimulating factor (in the presence of IL-4) for the production and secretion of YKL-39 in human primary macrophages in the late stages of their differentiation. YKL-39 was shown to have a very strong chemotactic effect on primary human monocytes. Moreover, it efficiently stimulated angiogenesis in vitro. Furthermore, it was found that elevated levels of YKL-39 in the tumor mass after the neoadjuvant chemotherapy (NAC) have a positive correlation not only with poor response to NAC but also with the increased risk of distant metastasis [36].

#### 3.3.2. (B) Non-Cytokine-Related Factors

##### Stage of Cancer

It was found that CD14+CD16+ monocyte level is directly correlated with the early stage of BC. No significant association was shown in correlating menstruation, lymph node metastasis, estrogen receptor (ER), progesterone receptor (PR), and human epidermal growth factor 2 (HER2) with the level of CD14+CD16+ monocytes. However, higher levels of CD14+CD16+ monocytes were shown in early-stage BC patients, especially those with stage I and/or small tumor size (T1–T2) in comparison with stage II-IV patients [33]. This finding suggests that further studies are needed to know the reason explaining the expansion of CD14+CD16+ monocytes in the early stages of BC. Our review suggests that this might have happened as a preliminary step for TAM differentiation in later stages.

##### BC Subtype

A study showed that monocytes actively proliferate in the TNBC, but not in the luminal tumors. This finding was supported by using the Ki67 proliferation marker. Primary human monocytes were co-cultured with five different BC cell lines; three cell lines were TN (MDA-MB-231, MDA-MB-468, and SUM-159) and two cell lines were luminal A (MCF-7 and T47D). Upon staining with the apoptosis markers annexin V and 7AAD, the TNBC culture medium showed a significant increase in the survival of monocytes in comparison to the luminal A culture medium. However, it was also noticed that a conditioned medium from a third TN cell line (MDA-MB-468 cells) did not induce monocyte proliferation [31]. Luminal A cell lines’ (MCF-7 and T47D) chemokine expression pattern, angiogenesis, and invasion-related proteins were analyzed in co-culture with monocytes. Consequently, it was observed that they express fewer factors in comparison with the TNBC cell lines. In this context, granulocyte macrophage colony-stimulating factor (GM-CSF), matrix metalloproteinase (MMP), endothelin-1, platelet factor 4 (CXCL4), IL-8, and C-C motif chemokine ligand 2 (CCL2) were more upregulated in TNBC cell line/monocyte co-cultures than in luminal A/monocyte co-cultures [31]. Thus, the TN tumor cell environment, alone or in co-culture with monocytes, contains important factors that maintain myeloid cell survival, proliferation, and differentiation [31]. Another study group supported this finding by co-culturing MDA-MB-231 with U937 (a pro-monocytic human cell line), and a significant increase in invasion and migration of U937 was recognized. These data support the finding that hormone-independent BC cell lines recruit more monocytes than hormone-dependent BC cell lines. Moreover, this study showed that this happens due to the secretion of larger amounts of CSF-1 in the hormone-independent BC using the MDA-MB-231 cell line rather than the hormone-dependent BC using the MCF-7 cell line [1].

However, in another study, the frequency of CD14+CD16+ monocytes was increased by MCF-conditioned medium (CM) and monocyte co-cultivation. Human PBMC-purified CD14+ monocytes were cultured in complete RPMI-1640 with or without varying doses of MCF-CM for 24 h [33]. The proportion of CD14+CD16+ monocytes was increased by almost 3-fold in the presence of 25% MCF-CM and by almost 5-fold in 50% and 75% MCF-CM. These results show that MCF-CM can significantly expand the level of CD14+CD16+ monocytes in a dose-dependent manner [33]. Monocyte chemoattractant protein-1 (MCP-1), also known as CCL2, is found to be a key player in the monocytes’ expansion in MCF-CM. MCP-1 is one of the chemokines produced by the immune cells, mainly monocytes and macrophages, and is overexpressed in breast tumor cells and some other tumor cells [37]. On detecting its levels by ELISA, the level of MCP-1 in monocytes treated with MCF-CM for 24 h was higher than that in controls. After the use of neutralizing antibodies against MCP-1, the expansion of CD14+CD16+ monocytes by MCF-CM was inhibited. These findings suggest that CD14+CD16+ monocytes increased in patients with BC via the stimulation of monocytes to secrete more MCP-1. This stimulation is triggered by MCF-CM. On the secretion of MCP-1, more monocytes are recruited to the TME [33]. Some studies relate the expansion of CD14+CD16+ to TNBC, but others suggest that any BC subtype will expand the monocytes effectively. We suggest that CD14+CD16+ expansion can happen with any BC cell line but may happen with its highest levels with TNBC cell lines.

### 3.4. Monocytes’ Differentiation 

#### 3.4.1. Monocytes’ Differentiation to Dendritic Cells

Monocytes are plastic by nature; they can differentiate into macrophages and dendritic cells [38]. CD14+CD16+ monocytes are found to be more similar to DCs than CD14+ CD16− monocytes. They show a higher potential to differentiate into DCs than CD14+CD16− monocytes in a model of transendothelial trafficking [33]. Thus, CD14+CD16+ monocyte-derived DCs (CD16+ mDCs) help in the activation of T helper type 2 (Th2) responses when compared to CD14+CD16− mDCs [33]. 

#### 3.4.2. Monocytes’ Differentiation to MDSCs

Myeloid-derived suppressor cells (MDSCs; CD34+CD33+CD13+CD15(−)) are a population of myeloid progenitor cells that have been observed to be key players in chronic inflammation and cancer development. MDSCs promote tumor cell growth and suppress immune cell function through various mechanisms. One of these mechanisms is the production of arginase 1 (ARG1), which firstly contributes to inducible nitric oxide synthase (iNOS) to increase levels of superoxide and nitric oxide (NO). NO in its turn reduces lymphocyte function. Secondly, it can contribute to T-cell suppression, reinforcing the immunosuppressive ability of MDSCs [39]. MDSCs are found in TME of BC patients with very high levels compared to those in normal patients. MDSCs include granulocytes and immature cells which lack the expression markers for fully differentiated monocytes or granulocytes. In addition to the previous immune suppression mechanisms, MDSCs isolated from BC tissues show high expression of indoleamine 2,3 dioxygenase (IDO) [27]. IDO is an enzyme responsible for the catabolism of tryptophan. On depletion of tryptophan by IDO in the TME, kynurenine-based by-products are produced, leading to the inhibition of T-cell proliferation and promoting T-cell apoptosis [27].

#### 3.4.3. Monocytes’ Differentiation to Macrophages 

##### Monocytes’ Differentiation to M1 Population (Pro-Inflammatory) 

Monocytes can differentiate into M1 macrophages, which express the CD163 receptor marker in low levels (CD163^neg/low^ macrophages) and mediate defense immune response against bacterial pathogens [1]. M1 macrophages usually receive stimulation from GM-CSF, LPS, and IFN-γ to produce IL-23 and IL-12 and promote Th1 responses [40], thus having a pro-inflammatory response. In addition to this, M1 can secrete IL-6, ROS, and TNF-α [41]. Indeed, in vitro generation of M1-like macrophages can be accomplished by the stimulation from GM-CSF, LPS, and IFN-γ [42].

##### Monocytes’ Differentiation to M2 Population/TAMs (Anti-Inflammatory) 

Mechanism of Activity

Monocytes have the potential to differentiate into the M2 phenotype, which is CD163^high^ [33]. The M2 phenotype can reduce tissue damage caused by inflammatory processes and stimulate their repair, thus having anti-inflammatory responses. M2 is usually activated by M-CSF, IL-4, IL-10, and IL-13 and can produce anti-inflammatory IL-10 and TGF-β. In a research study, M2 is classified into three subgroups: M2a, M2b, and M2c. M2a is differentiated upon the stimulation by IL-13 and IL-4 producing matrix remodeling cytokines. M2a plays an important role in wound healing, thus acting as an anti-inflammatory immune cell with an elevation in the expression of both CD200R and CD86. Additionally, the M2b subgroup is stimulated by IL-1β or LPS, also displaying immunosuppressive activity. Moreover, M2c is stimulated by IL-10 and TGF-β. It is of interest to know that M2c macrophages generate further IL-10 and MMPs with a significant elevation in CD163 expression [35].

In the presence of tumors, macrophages are plastic and can be reprogrammed to polarize into M1-like macrophages or M2-like macrophages [42]. They respond to the TME and the environmental stress in solid tumors [3]. Macrophages found in TME can be identified as CD45+CD11b+HLA-DR+CD14+BDCA1^neg^CD64+cells accounting for about 25% of living CD45+ cells [43]. Macrophages form two subgroups, namely CD163^neg/low^ and CD163^high^. CD163^neg/low^ resembles M1, having anti-tumor properties, and accounts for 15%. However, CD163^high^ resembles M2, having immune-suppressing, pro-tumorigenic, and pro-angiogenesis properties, and accounts for 9.7% [21]. Monocytes infiltrate tumor tissue and polarize to TAMs [3]. It is of interest to know that TAMs contribute to about 5–40% of the tumor mass in solid tumors [41]. 

In the beginning, TAMs are said to have a pro-inflammatory phenotype (M1-like) and inhibit tumorigenesis by ROS and TNF-α or even by phagocytosis [3]. However, in the late stages of cancer, TAMs start to secrete IL-10, TGF-β, and IL-12 as previously mentioned, suppressing cytotoxic T lymphocyte (CTL) and NK cells and thus having anti-inflammatory action [3]. Therefore, TAMs reduce the survival in BC patients, worsening their clinical outcomes [44], due to their anti-inflammatory activity that is performed through various mechanisms (Figure 1).

In vitro generation of M2-like macrophages can be achieved by the stimulation of monocytes by M-CSF, IL-4, and IL-10 [42]. Upon addition of a tumor-conditioned medium (TCM) to an anti-tumor cytokine cocktail (IL-4 and IL-10) in addition to M-CSF, TAMs can be generated [42]. As a matter of a fact, IL-4 and IL-10 inhibit pro-inflammatory cytokines’ production [11]. Furthermore, a TCM is important for the generation of TAMs as it contains some soluble factors that have a major role in the recruitment and the adhesion of myeloid cells to tumor cells. These factors are VEGF-A, IL-8, IFN- γ, G-CSF, and macrophage inflammatory protein-1b (MIP-1B) [46]. Indeed, on differentiation of monocytes into TAMs in vitro, TAMs are found to be phenotypically different from the traditional M2- and M1-like macrophages [48]. It was found that CD14 is expressed at higher levels in M2-like macrophages than in M1-like macrophages. TAMs express CD14 at lower levels than both M1- and M2-like macrophages. However, TAMs co-express CD163/206 at a higher level than M1- and M2-like macrophages, highlighting that TAMs differ from M2 due to the presence of the tumor [42]. In another study, the U937 cell line was differentiated into TAMs. This research group used specific surface antigens CD163 and CD204 as evidence of differentiation [2]. Thus, collectively, CD163, C206, and CD204 are considered self-markers for TAMs. 

Besides these markers, TAMs express markers for immune suppression, invasion, and angiogenesis at higher levels than M1- and M2-like macrophages. In addition, they express IL-10, IL-8, CCL2 (MCP-1), CCL7 (MCP3), macrophage inflammatory protein (MIP-1B), and VEGF-A at higher levels than M1- and M2-like macrophages. As mentioned before, IL-10 is well known for its immune-suppressive action that contributes to T-cell regulation [42]. MCP-1 is a monocyte recruiter and drives metastasis and growth to the lung and bone in nude mice [46]. As for IL-8, it has been shown to promote tumor cell invasion activity in a human colon cancer cell line [49]. However, the expression levels of IFN-γ, IL-6, G-CSF, IL-13, IL-17A, TNF-α, and IL-1β were lower in TAMs than in M1- and M2-like macrophages. M2 macrophages expressed IL-13 at the highest levels. The highest level of IL-6 expression was observed in M1 macrophages [42]. Monocytes are known to enhance NK cell cytokine response through the interaction between natural killer group 2D (NKG2D) receptor on NK cells and MHC class I polypeptide-related sequence A (MICA) on the monocytes. Accordingly, it is important to observe the effect of monocyte-derived macrophages on NK cells. NK cells are well known to be key players in immunosurveillance. On co-culturing M1-like macrophages with NK cells, they were shown to enhance the NK response in comparison to M2-like macrophages and TAMs. M2 and TAMs were shown to suppress IFN-γ production by NK cells [42]. 

In addition to this, IL-8 production by M2 macrophages can be induced by leptin. Leptin is produced by adipose tissue; as a consequence, when the adipose tissue mass increases, the levels of leptin increase [50]. It was reported that high levels of leptin and tumor aggressiveness are correlated [50]. This may explain why obesity is a BC risk factor. Leptin binds to the leptin receptor complex (ObR) to activate canonical signaling pathways (JAK2/STATs, PI-3K/AKT, MAPK/ERK 1/2) as well as non-canonical signaling pathways (p38, JNK, PKC, MAPK, and AMPK) [51]. Leptin has a direct action on M2 macrophages by increasing their expression of the leptin receptor as well as stimulating them to express IL-8. IL-8 production was shown to be sensitive to ERK inhibitor PD980590, p38 MAPK inhibitor, and anti-ObR neutralizing antibody [38], which supports the idea that leptin interacts with its receptor to activate MAPK/ERK 1/2 and P38/MAPK signaling pathways in M2 macrophages. However, on testing the effect of the inhibitors of JAK (AG490: 50 μmol/L), PI3K (LY294002: 10 μmol/L), and JNK (SP600125: 50 μmol/L), they showed no significant effect on IL-8 expressed by M2 macrophages. Interestingly, leptin may induce the recruitment of macrophages to the TME by ER stimulation in the macrophages. In addition to this, it induces the macrophages to produce VEGF, IL-6, IL-1, and TNF-α [38].

##### Key Players Affecting the Differentiation of Monocytes to TAMs/M2

ACytokine-Related Factors

Interleukin 10 (IL-10)

As mentioned before, IL-10, IL-4, and M-CSF are considered to be the major key players in the differentiation of monocytes in vitro. Besides the role of IL-10 in the differentiation, it is of great importance to highlight the effect of IL-10 on the monocytes in the TME. Initially, IL-10 is known as a cytokine synthesis inhibitory factor inhibiting the pro-inflammatory gene expression in the T cells and macrophages. Furthermore, it inhibits the capacity of antigen-presenting cells, thus acting as an anti-inflammatory cytokine. IL-10 suppresses the function of T cells by suppressing the expression of IL-1α, -1β, -6, -8, -12, and -18; TNF-α; and GM-CSF in T cells. In addition, it suppresses the production of IFN-γ in activated T helper (Th) cells [52]. Moreover, IL-10 inhibits the expression of MHC class II expression with the upregulation of both surface proteins CD80 and CD86. It is important to also highlight that IL-10 inhibits the DC maturation and differentiation from monocytes. Generally, IL-10 binds to transmembrane receptor complex IL-10R, which is composed of two different chains: IL-10R1 and IL-10R2. IL-10R1 is the chain that determines cellular responsiveness. It was found that the IL-10R1 on the monocyte surface has 720 IL-10 binding sites making the monocytes express the highest IL-10R1 levels. Upon differentiation of monocytes to macrophages or DCs, the level of IL-10R1 was assessed. DCs were observed to downregulate the IL-10R1 expression levels and thus succeed in escaping the inhibitory action that is exerted by IL-10 on the DC population. On the contrary, the differentiated anti-inflammatory macrophages upregulate the IL-10R1 on their surfaces [53]. Besides the role of IL-10 in polarizing the monocytes to the anti-inflammatory M2/TAMs, it can also enhance the monocytes’ anti-inflammatory actions through various mechanisms; thus, the increase in the levels of IL-10 by different stimulators is expected, as shown in Figure 2. 

Monocyte Chemoattractant Protein-1 (MCP-1/CCL 2)

In addition to its previously mentioned function as a monocyte chemoattractant, MCP-1 has been found to have a potential capability to shift the differentiation of monocytes to TAMs [33] that increase tumor tolerance [1].

TGF-β, VEGF

In a research study, CD14+ monocytes were first isolated from healthy donors’ blood. Then, they were cultured for 7 days in the presence of supernatant from primary dilacerated tumors, and the supernatants were named SNDils. SNDils are supernatants of breast tumor tissues prepared by the mechanical dissociation of the tumor in a culture medium. After leaving the monocytes for 24 h of LPS activation, levels of cytokines and surface molecules were assessed to observe the differentiated population, which was named SNDil macrophage (SNDil-MΦ). The SNDil-MΦ population was compared to the control populations, which were different subpopulations of macrophages (MΦs), namely M0-MΦ, M1-MΦ, and M2-MΦ, and a monocytic dendritic cell (Mo-DC) population. Each population was generated by the differentiation of CD14+ monocytes in vitro under well-defined conditions [43].

Every macrophage subpopulation (M0-MΦ, M1-MΦ, M2-MΦ, and SNDil-MΦ) or Mo-DCs was shown to have its own unique set of surface markers (Table 1). All macrophage subpopulations (M0-MΦ, M1-MΦ, M2-MΦ, and SNDil-MΦ) were shown to be CD14+CD64+BDCA1^low^, while Mo-DCs were CD14^low^CD64^neg^BDCA-1^high^. On comparing the macrophage subpopulations with each other, CD163 was shown to be expressed at the highest levels in M2-MΦ, while it was lost completely in M1-MΦ. As for the SNDil-MΦ population, CD163 showed heterogeneous levels and was classified into two groups: CD163^high^ SNDils, which were similar to M2-MΦ and made up to 51% of all SNDil-MΦs, and CD163^neg/low^ SNDil-MΦs, which were very similar to M0-MΦ [43]. There was a positive correlation reported between CD163 and IL-10 levels. Thus, IL-10 produced by CD163^high^ SNDil-MΦ was significantly higher than CD163^neg/low^ SNDil-MΦ. Autocrine IL-10 production plays a very major role in the upregulation of CD163 and PD-L1. This finding was observed upon the addition of neutralizing anti-IL-10Ra mAb to the M2-MΦ population, with no reported change in the levels of PD-L2, CD80, or CD86. Additionally, the major cytokines and chemokines in the TME responsible for the differentiation to M2-like SNDil-MΦ (CD163^high^IL-10^high^ MΦs) were investigated. Accordingly, levels of CCL2 (MCP-1), M-CSF, TGF-β1, TGF-β3, and VEGF were shown to be significantly higher than those for CD163^low^IL-10^low^ MΦ [43]. To confirm this finding, antibodies neutralizing M-CSF, pan TGF-β, and VEGF were added during differentiation. It was observed that levels of CD163, PD-L1, and IL-10 induction were impaired; however, the level of CD86 was increased. Indeed, CD86 was found to be higher in M1-MΦ than in M2-MΦ [43]. This study shows that TGF-β, M-CSF, and VEGF in the TME polarize the monocytes towards the CD163^high^CD86^low^IL-10^high^ MΦ population (SNDil-MΦ population), which is described in other studies as the tumor-associated macrophage (TAM) population.

BNon-Cytokine-Related Factors

BC Subtype

One study showed that U937 cell line polarization towards TAMs is influenced by hormone-independent BC cells compared to hormone-dependent cell lines. The detection of CD163-positive cell expression levels by immunohistochemistry showed that they were significantly higher in hormone-independent BC. Fifty percent of hormone-independent tumor volume was observed to be made up of TAMs. This finding has no significant correlation with tumor size, HER2 expression, lymph node involvement, proliferation index (Ki67), or menstrual status. CD204 was also assessed and was found to be upregulated. Generally, levels of CD163+ and CD204+ macrophages were high with poor overall survival [1].

In the same study, a CSF-1-overexpressing MCF-7 stable cell line was generated using lentivirus particles. After generation, the CSF-1-overexpressing MCF-7 cell line was co-cultured with U937 cells to show its potential in monocyte differentiation. CSF-1-overexpressing MCF-7 stable cells failed to differentiate U937 cells into M2-like macrophages. Generally, on differentiation of monocytes to M2 macrophages, some characteristics that ensure differentiation are observed, such as morphology, large cell volume, increase in expression of surface antigens CD163 and CD204, and increase in adherence. On analyzing U937 cells co-cultured with the CSF-1-overexpressing MCF-7 cell line for the same characteristics, nothing was observed. Additionally, TAM differentiation and infiltration were abrogated by knocking down CSF-1 in the MDA-MB-231 cell line [1]. 

In another research study, key players behind the attractive medium of MDA-MB 231 for M2 (TAM) differentiation were investigated in comparison to other conditioned media. It was found that M-CSF is the major factor secreted at very high levels from this specific cell line [41]. In this context, we conclude that the hormone-independent cell lines might have a greater potential in the differentiation of monocytes to TAMs/M2 than hormone-dependent cell lines due to the presence of critical key players such as M-CSF.

Phorbol-12-Myristate-13-Acetate (PMA)

In a study, PMA (100 nM) with the presence of IL-4 was shown to help THP1 (human leukemic monocyte cell line) differentiate into the M2 phenotype [38]. Thus, PMA is a potential player in the in vitro differentiation of monocytes to the anti-inflammatory macrophage population.

Noncoding RNAs 

While protein-coding genes account for about 2% of the genome, more than 98% of the genome is noncoding, consisting of small interfering RNA (siRNA), small RNA (miRNA), and long noncoding RNA (lncRNA). Long noncoding RNA is made up of more than 200 nucleotides [58]. lncRNAs are classified according to genomic location into four major classes; the largest group is long intergenic lncRNAs (lincRNAs) and is a key player in the cell cycle, gene expression, and immune cell differentiation [59]. It was also shown that this group contributes to TAM polarization. One of the well-known lincRNAs is lincRNA-p21.

Initially, lincRNA-p21 regulates the expression of p53, which is one of the key players in suppressing tumor progression by regulating DNA damage and cell cycle arrest. Moreover, some research data showed that lincRNA-p21 is decreased in some types of tumors and inflammatory diseases. Interestingly, lincRNA-p21 was shown to have a role in the polarization of TAMs. It was found that lincRNA-p21 is upregulated significantly in TAMs, and its knockdown reverses the phenotype of TAMs, resulting in the upregulation of CD86 and downregulation of CD206 (as previously mentioned, CD206 is a self-marker for TAMs/M2). It is proposed that this happens due to its regulatory activity on p53. Additionally, on knocking down lincRNA-p21, anti-tumor cytokines IL-6 and TNF-α were increased while the anti-inflammatory cytokines IL-4 and IL-10 were inhibited. Moreover, among TAMs in which lincRNA-p21 was downregulated, tumor cell apoptosis was promoted while migration and invasion were inhibited. These findings suggest that lincRNA-p21 knockdown facilitates the polarization into M1-like macrophages, increasing the M1 macrophage proportion and decreasing the M2 macrophage proportion, and thus has a pro-inflammatory anti-tumor action [3].

Besides lncRNAs, miRNAs are also known in previous studies to have a major role in the differentiation of monocytes into macrophages. Accordingly, it was previously observed that miR-223 has a major role in differentiation by targeting a cell cycle regulator, E2F1, and thus causing an exit from the cell cycle [60]. On the contrary, the upregulation of miR-22 in monocytes promotes their differentiation through c-JUN expression and its interaction with PU.1, which controls the whole differentiation process [61]. It is also worth mentioning that elevated levels of miR-424-5p, -362-3p, -335-5p, and miR-106 were observed in macrophages in comparison to progenitor cells, meaning that they could possess a major role in differentiation [62]. In addition, miRNAs showed a role in macrophage recruitment through the expression of high levels of miR-375, which enhances CCL2 (its role in recruitment was previously mentioned in this review) [63]. On the contrary, miR-125 was shown to be overexpressed in tumor cells, leading to the inhibition of M-CSF by tumors thus reduction in macrophage recruitment [62]. Additionally, it was shown that miRNAs have a role in metastasis. One example is miR-3607. Briefly, breast cancer metastasis is promoted by the lncRNA called circIRAK3, which sponges miR-3607. Commonly, miR-3607 causes the downregulation of forkhead box C1 (FOXC1). Accordingly, FOXC1 is upregulated upon the sponging of miR-3607 [64]. The exact mechanisms of activity for FOXC1 were previously discussed in the referenced review [65].

### 3.5. TAM Gene Signature

A study group analyzed 37 genes of TAMs and observed the highest expression of colony-stimulating factor 1 (CSF-1) response signature with shorter disease-specific survival (DSS). CSF-1 was shown to be expressed at higher levels in luminal B, HER2, basal, and claudin-low compared to luminal A [66].

One limitation in studying TAMs has been the lack of specific markers. Indeed, as mentioned previously in Figure 1, the most known marker for TAMs in the literature is CD163. However, it has also been reported that the SIGLEC1 gene encoding CD169 is among the top upregulated genes, and it was barely detectable in the circulating monocytes. Furthermore, higher numbers of CD163 and SIGLEC1 single- and double-positive TAMs were reported in invasive breast cancer. It is worth mentioning that a high SIGLEC1 expression level was independently associated with a shorter DSS, suggesting that SIGLEC1 is a TAM-restricted biomarker. The same study correlated SIFLEC1 with CCL8, suggesting that together they are significantly correlated with DSS. Thus, this study considered CCL8 as an additional biomarker for TAMs [67].

Another study performed single-cell RNA sequence data analysis to identify macrophage marker genes in breast cancer (MMGS). The Cancer Genome Atlas (TCGA) database was used to build the MMGS model as a training cohort. However, as a validation cohort, the GSE96058 dataset was used. A number of genes were included in the MMGS model, namely SERPINA1, CD74, STX11, ADAM9, NF-KB, and PGK1. MMGS risk score was correlated with overall patients’ survival and was divided accordingly into high- and low-risk groups. Results showed that low expression of SERPINA1 predicts patients’ poor outcome, indicating SERPINA1 has anti-neoplastic roles in breast cancer. Furthermore, syntaxin 11 (STX11) is reported to be overexpressed in monocytes/macrophages, NK cells, and T cells. Silencing of STX11 in macrophages enhanced the phagocytosis of apoptotic cells and antibody-dependent target cells and enhanced the secretion of TNF, showing that STX11 has anti-tumoral activity [68]. Furthermore, ADAM9 is found to be expressed by monocytes and activated macrophages. ADAM9 was shown to degrade ECM proteins, indicating its metastatic potential in tumor progression [69]. Additionally, NF-KB signaling is proposed in a study to be the central mechanism maintaining the immunosuppressive phenotype of TAMs. [70]. Last but not least, this study also suggests that PGK1 might participant in macrophage activation; however, this needs further investigation [71].

### 3.6. Role of TAMs and MDSCs in BC Metastasis

Metastasis is explained in some papers by the “seed and soil” theory, in which the cancer cell is considered the seed while the TME is the soil. Cancer cells undergo epithelial-to-mesenchymal transition (EMT), thus invading the blood vessels and becoming circulating tumor cells (CTCs) which eventually colonize as disseminating tumor cells (DTCs) [41]. TAMs (M2 phenotype), as key players in the TME, were shown to be recruited in the hypoxic areas. This might be an explanation for why macrophages in the early stages of colon, lung, and stomach cancers having normoxic milieu polarize more to the M1 phenotype, thus resulting in patients having a good prognosis [41]. However, it is very important to state that a study showed a negative correlation between progression and TAM abundance in specific intratumoral areas with lymphatic metastasis. This might suggest that some subpopulations of TAMs can regain anti-tumor properties in human BC [36]. However, most of the studies showed that TAMs have metastatic potential that interferes with all the steps of metastasis. Generally, anti-inflammatory monocyte-derived populations (TAMs and MDSCs) contribute to invasion, intravasation, and creation of a metastatic niche (Figure 3).

#### 3.6.1. Monocytes’ Differentiation to M0 Macrophages 

M0 macrophages are derived from the bone marrow and are known to be resting-state macrophages. They are usually considered precursors of polarized macrophages, either M1 or M2, without a specific function before their polarization. However, a recent study compared the immunophenotyping of glioma-associated macrophages versus matched blood monocytes, healthy donor monocytes, normal brain microglia, non-polarized M0 macrophages, and M1 and M2 macrophages. It has highlighted that macrophages that infiltrate into glioma tissues resemble M0 macrophages. Interestingly, further analysis of glioma data from The Cancer Genome Atlas (TCGA) and the Chinese Glioma Genome Atlas databases confirmed that differentiation of M0-like macrophages, rather than M1 or M2 macrophages, is associated with a poor prognosis and high-grade tumor in glioma [75].

#### 3.6.2. Monocytes’ Differentiation to Mregs

Human Mregs suppress mitogen-stimulated T-cell proliferation in vitro through interferon-gamma (IFN-γ)-induced indoleamine 2,3-dioxygenase (IDO) activity. Additionally, Mregs induce the development of activated Tregs that, in turn, suppress the proliferation and activity of effector T cells. It was observed that Mregs are derived from CD14+ peripheral blood monocytes upon culturing in the presence of M-CSF and high concentrations of heat-inactivated human serum for more than 4 days before stimulation with IFN-γ [76] Moreover, DHRS9 was identified as a specific and stable marker for Mregs. Furthermore, low expression of CD163, CD14, and CD16 distinguished human Mregs from the rest of the human monocyte-derived macrophages [77].

#### 3.6.3. Monocytes’ Differentiation to M3 

M3 TAMs are also described as M1/M2 or M2/M1 switch phenotype both in mice and humans. Indeed, the majority of TAMs observed were M2-polarized phenotype (63%), whereas there were smaller numbers of M0 (21%), M1 (7%), and M3 (9%) phenotypes. As M3 is known to be a switch group, a combination of the human M1 (iNOS) and M2 (CD163) markers was used to identify M3 macrophages (iNOS+CD163+) [78]. M3 TAMs were also reported to have anti-tumor activity in Ehrlich ascites [79] and a prostate cancer mouse model [80]. 

#### 3.6.4. Monocytes’ Differentiation to M4

In another study performed in the context of atherosclerosis, M4 macrophages were identified as a unique subset of MCSF/CXCL4-dependent macrophages. M4 macrophages expressed high levels of a wide range of genes, namely CD86, TNF superfamily member 10 (Tnfsf10), mannose receptor c type 1 (Mrc1), CCL18, and CCL22. However, it expressed lower levels of pentraxin 3 (Ptx3), CD36, and IL-10. Surprisingly, M4 was shown to be weakly phagocytic in a study performed within atherosclerotic plaques [81]. In this review, we recommend more investigation into the role of M4 in tumors.

#### 3.6.5. Monocytes’ Differentiation to Mox

A unique subtype of macrophages was generated upon the treatment of unpolarized bone marrow-derived macrophage cultures with ox-PL 1-palmitoyl 2arachidonoyl-sn-glycero-3-phosphorylcholine. Accordingly, this subset was named Mox macrophages. Mox expressed a unique profile of genes including heme oxygenase-1 (HO-1), sulfiredoxin-1 (Srnx1), and thioredoxin reductase 1 (Txnrd1) in a nuclear factor erythroid-derived 2-like 2 (Nrf2)-dependent manner [82]. The proposed phenotype of Mox macrophages closely resembles the phenotype of the recently proposed hemorrhage-associated macrophages (Mhas) [83]. At present, little is known about the functionality of Mox in vivo. However, Mox expresses IL-10, VEGF, and enzymes with anti-oxidizing activities; thus, it may exert anti-inflammatory actions in vivo. In addition, Mox-derived IL-10 and VEGF may play anti-inflammatory roles through the suppression of T cells and the promotion of endothelial cell proliferation and survival, respectively. It is worth mentioning that data from in vitro phagocytosis assays demonstrated that Mox macrophages have very poor phagocytic activity [82].

#### 3.6.6. Monocytes’ Differentiation to Hemorrhage-Associated Macrophages (Mhas) 

It was observed in a number of studies that Mox macrophages seem to be closely related to the recently identified hemorrhage-associated macrophages (Mhas). Monocytes can be differentiated to Mhas in vitro by using hapto-hemoglobin complexes or oxidized red blood cells leading to the upregulation of CD163 and IL-10. Generally, recently identified macrophage subsets (M4, Mox, and Mha) are still poorly characterized and studied in cancers [84].

#### 3.6.7. Monocytes’ Differentiation to M17

A research study proposed that the differentiation of monocytes to M2c phenotype depends mainly on the type of T-helper cytokine environment besides the stage of macrophage differentiation. Briefly, the presence of IFNγ (Th1 inflammation) or prolonged exposure to IL-4 (Th2 inflammation) causes resistance to M2c differentiation, thus resulting in impaired clearance of apoptotic neutrophils, uncontrolled accumulation of apoptotic cells, and persistent inflammation. However, the presence of IL-17 (Th17 environment) prevents monocyte-derived macrophage apoptosis and leads to intense M2c differentiation, thus ensuring efficient clearance of apoptotic neutrophils and restoring anti-inflammatory activity. The data presented in this study support an unexpected role for IL-17 in orchestrating the resolution of innate inflammation; moreover, the macrophage population that was polarized to M2c in the presence of IL-17 was named M17 [85]. In this review, we do not support this extensive categorization as it is not clear enough in other research studies whether the rest of the populations are affected as well by IL-17 or not; thus, we propose that the impact of IL-17 on this population is not a sufficient basis for naming the population M17. Collectively, monocytes can differentiate into many phenotypes of macrophages, which are summarized in Table 2.

Our review aims to shed light on the importance of monocytes and their derived populations in the TME of breast cancer. The review also aims to highlight the behavior of the monocyte-derived populations, especially the anti-inflammatory populations. We suggest that studying the factors behind monocyte expansion and differentiation into anti-inflammatory populations (e.g., TAMs) could be a game changer in the immunotherapy research field.

## 4. Conclusions

In 2020, 685,000 new death cases were correlated to BC despite the presence of many therapies targeting BC tumors. Interestingly, targeting the TME and not the tumor itself could be a new treatment strategy. The TME of BC is composed of both pro-inflammatory and anti-inflammatory populations, namely CAFs that act as anti-inflammatory stromal cells secreting various growth factors and cytokines. Consequently, it promotes proliferation, angiogenesis, ECM remodeling, and metastasis. Moreover, tumor-associated neutrophils (TANs) are also found in TME and can be classified into the N1 phenotype having pro-inflammatory activity or the N2 phenotype having anti-inflammatory activity according to the presence of cytokines in the TME. This anti-inflammatory activity of TANs is suggested to happen via oncostatin M and is more common in TNBC compared to hormonal subtypes. Additionally, tumor-associated eosinophils possess pro-inflammatory activity, and their low counts are associated with poor prognosis. In addition to these TME components, monocytes are key players in the TME and can further polarize to different populations such as MDSCs (CD34+CD33+CD13+CD15(−)) known for their pro-tumorigenic activity. This anti-inflammatory activity happens through the suppression of T cells via the expression of a high level of ARG1 or IDO. Monocytes can also polarize to various phenotypes of macrophages; however, in this review, we focus mainly on TAMs. TAMs have two phenotypes: the M1 phenotype, which is pro-inflammatory and is known to be present in the early stages of BC before progression, and the M2 phenotype, which exhibits anti-inflammatory activity through various mechanisms and becomes abundant once cancer starts progressing (Figure 1). Due to the importance of monocytes, it is also important to study the factors that contribute to their expansion. These factors are divided in our review into cytokine-related factors and non-cytokine-related factors. The cytokine-related factors are M-CSF, CXCL16, and YKL-39; the non-cytokine-related factors are the stage and subtype of BC.

As a matter of a fact, TAMs are shown to be the most important population polarized from monocytes. This polarization is also influenced by cytokine-related factors, namely IL-10, MCP-1, TGF-B, and VEGF. IL-10 not only enhances the polarization towards TAMs (M2 phenotype) but also suppresses T-cell activity and the production of some pro-inflammatory cytokines. IL-10 is influenced by a number of stimulators, as shown in Figure 2. Moreover, MCP-1 is shown to possess the potential to enhance M2 polarization. In addition, TGF-B and VEGF in the TME also contribute to M2 polarization. On the other hand, this differentiation is influenced by non-cytokine-related factors, namely BC subtype, and epigenetically by a number of noncoding RNAs.

This review recommends additional research to fully understand the nature of monocytes in the TME and explore the different factors affecting their expansion and polarization. This knowledge in turn could be further used in the clinical implementation of BC treatment. Future studies should focus more on the genetic and epigenetic regulations behind the factors affecting the monocytes’ expansion and polarization. Thus, by controlling and limiting the polarization of monocytes to anti-inflammatory populations, the immunosuppressive activity may be significantly decreased, improving the prognosis and patient survival.

## Figures and Tables

**Figure 1 cancers-14-05460-f001:**
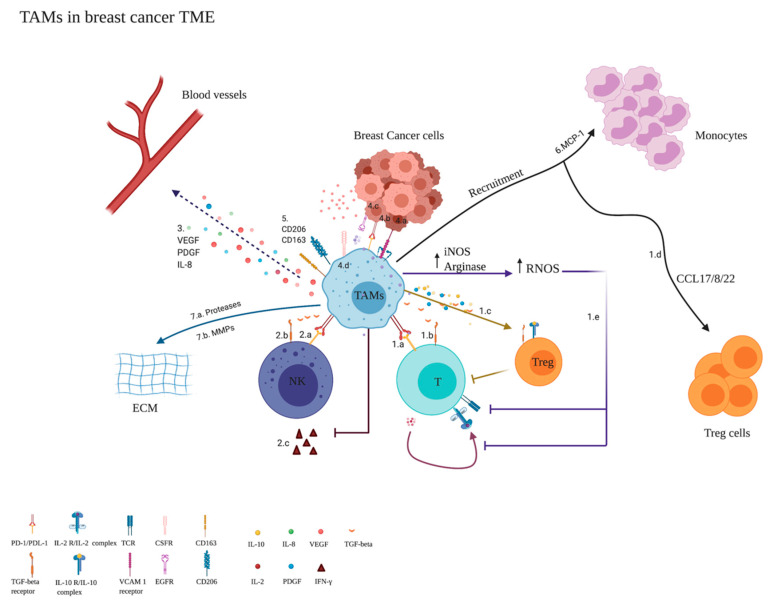
TAMs in the breast cancer TME. 1. The effect of TAMs on T cells: 1.a. TAMs can suppress T-cell proliferation via the programmed death-ligand 1 (PDL-1) [42]. 1.b. TGF-beta affects T-cell function by upregulation of PDL-1 on TAMs [42]. 1.c. Tregs are induced by IL-10, TGF-B, and PDGF-2, thus suppressing T cells [45]. 1.d. Treg recruitment happens through CCL7/8/22 [45]. 1.e. Increased activity of arginase enzyme and iNOS result in the increased level of NO and RNOS (ONOO^−^) leading to nitrosylation and thus impairing T-cell self-stimulation by IL-2 in addition to nitration of TCR signaling complex altering T-cell function [23]. 2. The effect of TAMs on NK cells: 2.a. The inhibitory effect of TAMs can be due to the expression of PDL-1 (which is highly expressed on TAMs) or 2.b. it can be due to TGF-B secretion [42]. 2.c. TAMs also suppress NK IFN-Y production [42]. 3. TAMs induce angiogenesis by releasing VEGF, PDGF, and IL-8 [23]. 4. Interaction between TAMs and cancer cells: 4.a. It was observed that VCAM1+ tumor cells have increased survival in a leukocyte-rich environment due to the adhesion of leukocyte receptors on BC cells (VCAM1) to TAM α 4 integrin [21]. 4.b. TGF-B was shown to upregulate PDL-1 on cancer cells, thus inducing an inhibitory effect on immune cells [23]. 4.c. There is a paracrine loop between cancer cells and TAMs. TAMs secrete epidermal growth factor (EGF) that binds to EGFRs on the cancer cells [21]. 4.d. TAMs express M-CSFR, which is a monocyte colony-stimulating factor receptor also known as colony-stimulating factor 1 receptor (CSF-1R or cFMS). The M-CSFR binds to the M-CSF (CSF-1) that is produced by cancer cells [41]. The binding of cancer cells with TAMs allows the co-migration of two different cell types, thus enhancing invasion, motility, and intravasation [21]. 5. CD163 and C206 are considered the commonly used self-markers for TAMs. 6. MCP-1 is a monocyte recruiter that is produced by TAMs; monocytes respond to TME and differentiate into TAMs [46]. 7.a. TAMs coordinate in the extracellular proteolysis through the secretion of tissue-remodeling cysteine cathepsin proteases that contribute to ECM and collagen degradation [47]. 7.b. Moreover, MMPs contribute to collagen degradation. Types I, III, IV, and VI are the major collagens that play an important role in tumors [47]. Myeloid cells remodel ECM by degrading the collagen through matrix metalloproteinases (MMPs) [34].

**Figure 2 cancers-14-05460-f002:**
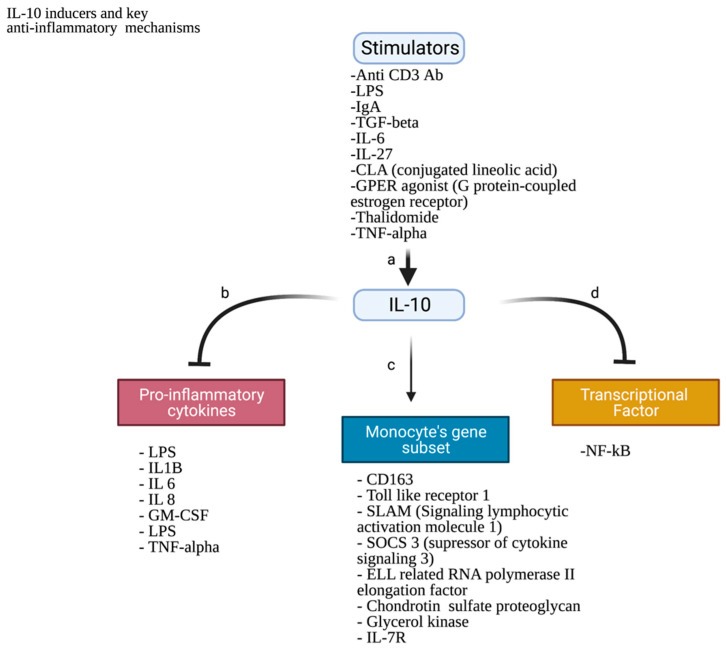
IL-10: key inducers and key anti-inflammatory mechanisms. a. Inducing agents increase the release of IL-10 from different immune cells [25,54,55] b. IL-10 acts as an inhibitor for a number of pro-inflammatory cytokines released by macrophages/monocytes. c. IL-10 induces a small subset of genes in human monocytes [56]. d. IL-10 is an inhibitor for NF-kB (nuclear localization of nuclear factor kB), a transcriptional factor responsible for the expression of inflammatory genes; however, other transcriptional factors such as NF-IL6, AP-1, and AP-2 were not affected by IL-10 [57].

**Figure 3 cancers-14-05460-f003:**
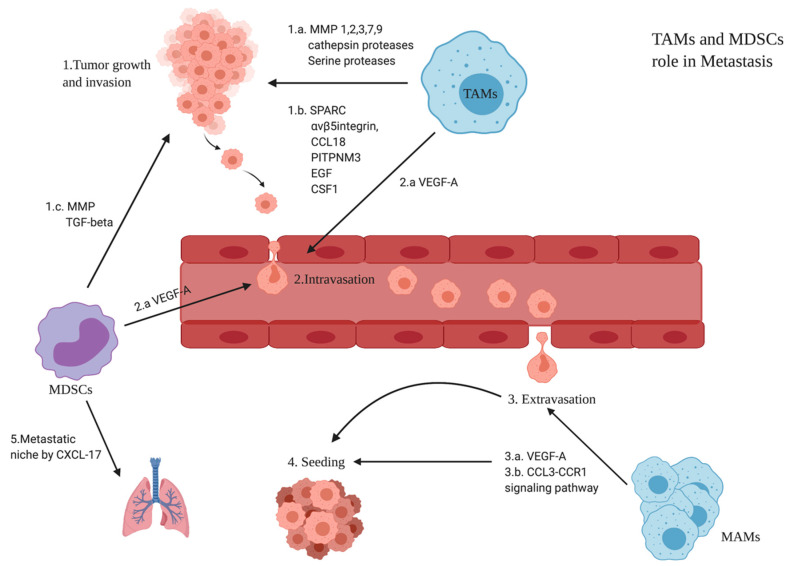
Role of TAMs and MDSCs in metastasis: 1. Tumor growth and invasion. 1.a. TAMs produce MMPs (MMP1/2/3/7/9) and cathepsin. 1.b. Chemokines such as acidic and rich in cysteine (SPARC), chemokine (C-C motif) ligand 18 (CCL18), αvβ5 integrins, phosphatidylinositol transfer protein 3 (PITPNM3), epidermal growth factor (EGF), EGF receptor (EGFR), colony-stimulating factor 1 (CSF-1), and CSF-1 receptor (CSF-1R) allow the interaction between tumor cells, thus facilitating the invasion step [72].1.c. MDSCs facilitate the invasion step by producing MMPs and TGF-beta [73]. 2. Intravasation. 2.a. TAMs and MDSCs release VEGF-A [72,73]. 3. Extravasation. 3.a. Metastasis-associated macrophages (MAMs) accumulate at the metastatic site and also release VEGF-A. 3.b. The CCL2-CCR2 signaling pathway activates the CCL3-CCR1 signaling pathway in MAMs, leading to their accumulation in the metastatic site and thus attracting more tumor cells and prolonging the process of seeding. 4. Seeding [72]. 5. The metastatic niche in the lungs is induced by MDSCs through the production of CXCL-17 which recruits more MDSCs in the lungs, leading to an increase in the levels of platelet-derived growth factor-beta (PDGF-beta) that will induce angiogenesis, thus creating favorable conditions for metastatic cells [74].

**Table 1 cancers-14-05460-t001:** Different surface markers in monocyte-derived subpopulations *.

	M0-MΦ	M1-MΦ	M2-MΦ	SnDil-MΦ	mo-DCs
CD14	+	+	+	+	−/lo
CD64	+	+	+	+	−
BDCA	−	−	−	−	+
CD163	−/lo	−	+	+/−	N/A
CD86	N/A	+	−	+/−	N/A

* In a research experiment, CD14+ monocytes were isolated and treated under well-defined conditions to polarize into different macrophage subpopulations (M0-MΦ, M1-MΦ, M2-MΦ, and SNDil-MΦ) or Mo-DCs having different subsets of markers; SNDil-MΦ is a population that was treated in the presence of a tumor and is also known as the TAM population [43].

**Table 2 cancers-14-05460-t002:** An overview of the monocyte-derived macrophage populations.

Macrophage Phenotype	Stimulation	Function	References
M0	Resting-state macrophages	Commonly, they are considered only as precursors to M1 or M2 phenotypes. However, they might have tumorigenic activity in glioma	[75]
M1	LPS and IFN-γ	IL-23, IL-12, and promotes Th1 responses. It can also secrete IL-6, ROS, and TNF-α	[40,41,42]
M2a	IL-13 and IL-4	Produce matrix remodeling cytokines. They are considered anti-inflammatory immune cells with an elevation in the expression of both CD200R and CD86	[35]
M2b	IL-1β or LPS	Immunosuppressive	[35]
M2c	IL-10 and TGF-β	Immunosuppressive activity is achieved by production of IL-10 and MMPs with a significant elevation in CD163 expression	[35]
M2d	Leukemia inhibitory factor, IL-6	Not clear	[86]
Mreg	IFN-Y	Suppress mitogen-stimulated T-cell proliferation in vitro through interferon-gamma (IFN-γ)-induced indoleamine 2,3-dioxygenase (IDO) activity. Activates Tregs that in turn suppress effector T cells	[76,77]
M3	Unknown	Known as TAMs with an M1/M2 or M2/M1 switch phenotype. It was reported to have anti-tumor activity in Ehrlich ascites and a prostate cancer mouse model	[78,79,80]
M4	CXCL4	Pro-inflammatory in the context of atherosclerosis	[81]
Mox	ox-PL 1-palmitoyl 2arachidonoyl-sn-glycero-3-phosphorylcholine (might be a strong stimulator in vitro)	Produces IL-10 and VEGF; thus, it might have anti-inflammatory/angiogenic activity	[82,83]
Mha	hapto-hemoglobin complexes or oxidized red blood cells in vitro	CD163 and IL-10 upregulation; thus, it might have anti-inflammatory activity	[84]
M17	IL-17	Anti-inflammatory activity through the polarization to M2c	[85]

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
