# Peer review of "The Monocyte, a Maestro in the Tumor Microenvironment (TME) of Breast Cancer"

_cancers, 2022, doi:10.3390/cancers14215460_

Round 1

Reviewer 1 Report

The review highlights the role of monocytes as a central component of tumor growth and breast cancer progression. The topic is interesting, and the authors provide a good perspective on tumor-monocyte cross-talk in breast cancer.  A major limitation is the redundancy in some sections where similar mechanisms of activation, differentiation, or polarization of monocytes are mentioned.  More specifically, the following issues need to be addressed to improve the manuscript:

1- In general, the manuscript is wordy with redundant statements throughout the text. A good example is section 1.6 where a more concise writing is required.

2- More surprising, 90% of the manuscript is part of the “introduction”, the remaining 10% representing the “method “and “conclusion” sections. This is obviously not adequate for the reader. The manuscript needs to be reorganized, and some sub-sections can be merged. A “good” introduction (shorter with a few paragraphs) needs to be included to lay the foundation and provide background information on what concept of monocytes has already been discussed in previous publications. 

3- In lines 113-115, it is stated: "Moreover, TGF-β secretion by tumor cells causes their migration to the TME initiating the trans-differentiation of fibroblasts to CAFs ". This can be confusing and deviates from the focus of the assigned section where TGFB derives from CAFs, not tumor cells. An independent paragraph could be added to elaborate the tumor-mediated TGFB effect on fibroblasts.

4- Section 1.5 includes cytokine (e.g., M-CSF) and non-cytokine (e.g., stage of cancer, BC subtypes) related factors. This section needs to be reorganized based on such categories. 

5- The authors heavily use M1 and M2 (like) to functionally classify macrophages. While this facilitates our understanding, it undermines the heterogeneity of monocytic cells which goes beyond our understanding from in vitro studies. 

6- In the first section of the introduction, the authors describe breast cancer subtypes. It would be relevant to emphasize on immune profile variations between and within breast cancer subtypes. 

Reviewer 2 Report

In this article, the roles of monocytes in the tumor microenvironment of breast cancer were summarized. It is an interesting, comprehensive, and well-written review. An minor concern is on the monocytes differentiation, more information on it would strengthen this manuscript. Macrophage can be categorized as activated (M1) and alternatively activated (M2) which can be divided into M2a, M2b, M2c, and M2d. Macrophage also can further be differentiated into Mox, M4, M(Hb), and Mhem subtypes by distinct factors or microenvironments. Author could consider providing an further detailed discussion, and a table and/or figure on these macrophage subtypes.

Reviewer 3 Report

TME in title of the article submitted by authors needs to revise by mentioning full name of TME, throughout the article many time authors has mentioned abbreviated form first then full name. Please revise accordingly. 

Authors has mentioned very less data about miRNA and it's role as prognostic biomarker for early detection of breast cancer. Better tabulate data related to miRNA, gene targets, next-generation sequencing reports, ontology etc in the revised form of manuscript. 

Overall many punctuation and typological errors so please take care in the revised submission. 

Round 2

Reviewer 1 Report

The revised version has much improved. Please include a column for references in table 2 (Table 2. An overview on the monocytic-derived macrophage populations.)

Author Response

Thanks for your valuable feedback. As per your kind recommendation, a references column has been added to table 2 in the revised manuscript.